# Is the Systemic Agency Capacity of Long-Term Care Organizations Enabling Person-Centered Care during the COVID-19 Pandemic? A Repeated Cross-Sectional Study of Organizational Resilience

**DOI:** 10.3390/ijerph19095045

**Published:** 2022-04-21

**Authors:** Holger Pfaff, Timo-Kolja Pförtner, Jane Banaszak-Holl, Yinhuan Hu, Kira Hower

**Affiliations:** 1Institute of Medical Sociology, Health Services Research, and Rehabilitation Science, Faculty of Human Sciences & Faculty of Medicine, University of Cologne, 50933 Koln, Germany; kira-isabel.hower@bmg.bund.de; 2Faculty of Human Sciences, Research Methods Division, University of Cologne, 50931 Koln, Germany; timo-kolja.pfoertner@uk-koeln.de; 3School of Public Health and Preventive Medicine, Monash University, Melbourne 3004, Australia; jane.banaszak-holl@monash.edu; 4School of Medicine and Health Management, Tongji Medical College, Huazhong University of Science and Technology, Wuhan 430030, China; hyh288@hotmail.com

**Keywords:** person-centered care, AGIL, organizational resilience, agency capacity, COVID-19, long-term care, collective agency, systemic agency

## Abstract

The COVID-19 pandemic has strained long-term care organization staff and placed new demands on them. This study examines the role of the general ability and power of a long-term care organization to act and react collectively as a social system, which is called systemic agency capacity, in safeguarding the provision of person-centered care during a crisis. The question of how the systemic agency capacity of long-term care organizations helps to ensure person-centered care during the pandemic is an open research question. We conducted a pooled cross-sectional study on long-term care organizations in Germany during the first and second waves of the pandemic (April 2020 and December 2020–January 2021). The sample consisted of 503 (first wave) and 294 leaders (second wave) of long-term care organizations. The top managers of these facilities were asked to report their perceptions of their facility’s agency capacity, measured by the AGIL scale, and the extent to which the facility provides person-centered care. We found a significant positive association between the leaders’ perceptions of systemic agency capacity and their perceptions of delivered person-centered care, which did not change over time. The results tentatively support the idea that fostering the systemic agency capacity of long-term care organizations facilitates their ability to provide quality routine care despite environmental shocks such as the COVID-19 pandemic.

## 1. Introduction

Person-centered care (PCC) is an important goal in long-term care organizations (LTCOs). Different studies have shown that PCC improves the well-being of both care recipients and caregivers [1,2]. PCC is central in the nursing home culture change movement to improve the well-being of long-term care recipients [2,3] and has been identified as a pillar of high-quality long-term care [4,5,6]. Although PCC has been defined in various ways [7,8], for this study, the key definition is that it encompasses connecting with care recipients “as unique individuals and recognizing that they have their own subjective experiences and preferences” [2] (p. 730).

There have been several studies on the organizational roots of PCC [9,10,11] and on identifying organizational determinants of PCC in nursing [2,11,12], but past research has primarily focused on specific and narrow measures of organizational processes, such as whether the organization provides access to electronic personal health records or whether there are clear PCC implementation plans [2]. There has been considerably less research on the organizational preconditions enabling PCC and even less research on these enabling and resilience-strengthening factors during times of environmental shocks such as the early periods of the COVID-19 pandemic. The aim of this study is to address this knowledge gap.

The pandemic has challenged LTCOs to manage two problems simultaneously. To fight the pandemic, LTCOs have had to cope with pandemic-related issues such as implementing prevention rules, visiting bans, quarantine zones, test procedures, vaccinations, and use of protective equipment [13,14,15,16,17]. Those providing outpatient care have also faced difficulties such as maintaining hygiene standards and distance requirements under challenging circumstances [13,14,16]. Beyond pandemic-related issues, LTCOs have had to cope with the normal demands of daily routines, ensuring high-quality PCC. All of these challenges are aggravated when LTCOs also have to manage problems such as staff shortages, remuneration, and restructuring [18,19]. Additionally, there is a conflict between isolating care recipients to cope with the pandemic and prevent infections and providing PCC [20]. Under these complex conditions, some LTCOs have had difficulties in providing high-quality care during the pandemic, while others have not [21].

We argue that this variation is partly due to differences in the general capacity of LTCOs to act coherently as united and goal-oriented collectives. If LTCOs possess this general capacity to act collectively, they are better able to maintain routine tasks such as PCC despite having to cope with extraordinary burdens such as the pandemic. 

From an action theory perspective, LTCOs are organized collectivities that can adapt to the pandemic. To respond to both the normal and unusual demands during the pandemic, organized collectivities require a certain amount of systemic agency capacity. We define systemic agency capacity as the capability of a collective system such as an LTCO to fulfill four system-critical functions: adaptation (A), goal attainment (G), integration (I), and latent pattern maintenance (L). Within the structural–functional theory, these four functions are called AGIL functions [22,23]. They have to be performed to make a collective system ready to speak, act, and survive [23]. The adaptation function is fulfilled when a collective system is able to adapt to new situations because of given resourcefulness (e.g., organizational slack) and because of flexible organizational structures and mindsets. The goal attainment function is fulfilled when a collective of people has the ability to define and set desired goals collectively, to monitor and control goal attainment, and to correct false goal-oriented strategies and actions [24,25]. The integration function is fulfilled when the parts of a collective system are closely connected through mutual trust, cohesiveness, and supportive networks [24,25,26,27]. The latent pattern maintenance function is achieved when the value and knowledge system as well as the belief system are maintained by institutionalization of value-based structures and are transferred to the next generation of members by socialization [23,28]. Systemic agency capacity can be understood as a higher-order function of these four AGIL functions that transforms a collective of people into a social system capable of (a) shaping action; (b) acting in the long term in an autonomous, self-organized, and autopoietic manner; and (c) surviving its members. 

We argue that organizations with a high systemic agency capacity perform well on nearly all organizational performance dimensions because this capacity represents something similar to a general fitness of an organization that makes it generally ready to act in different situations and with regard to different tasks [29]. Resilience management [30] requires as a precondition an organization that is able to decide and act as an inseparable social unit. Thus, systemic agency capacity fosters resilience during crises such as COVID-19 by enabling LTCO to maintain routine tasks such as PCC under difficult conditions. 

Some studies support this view indirectly because they show that goal attainment and social integration are important for implementing PCC. For example, one study showed that goal-oriented leadership [11] facilitates the implementation of PCC. Another study demonstrated that hospitals that promote a culture of goal setting have been more successful in realizing PCC than hospitals without such a culture [10]. Additionally, a study by Stanhope et al. showed that transformational leadership, which incorporates having a vision and fostering a strong team climate among subordinates, facilitates the implementation of PCC [31]. In addition, some researchers have found that social cohesion in a health care organization is conducive to PCC [11].

Thus, our hypothesis is that the systemic agency capacity of LTCOs, which is measured by the AGIL scale, enables nursing homes to maintain and preserve high-quality routine tasks such as PCC in the presence of additional pandemic burdens.

## 2. Materials and Methods

We used cross-sectional data from an online survey of long-term care managers from outpatient and inpatient nursing and palliative care organizations in Germany surveyed in April 2020 (first survey) and between December 2020 and January 2021 (second survey). 

### 2.1. Selection of Participants 

The contact information (email addresses) for the German facilities was obtained from a freely accessible data register on the internet. We contacted 4333 facilities by email, of which 3195 were registered as outpatient care services, 865 as inpatient care services, and 273 as hospices [32,33]. Only managers of a long-term care facility were eligible to participate in the survey. Other employees of a long-term care facility or managers of other facilities that do not provide long-term care were not allowed to participate in the survey.

### 2.2. Measures

We measured the systemic agency capacity of LTCOs using the AGIL scale, which is an additive scale standardized from 0 to 100 for analyses, with higher values indicating higher levels of systemic agency capacity. This scale is based on Parsons’ AGIL concept [23], which states that social systems have to fulfill four functions (adaptation, goal attainment, integration, and latency) to be able to act and survive. The AGIL scale was developed in previous studies [34,35] and specifically measures leaders’ perceptions of the capacity of their organizations to easily adapt to different situations (adaption; item 1), install effective processes (adaption, item 2), set and strive for collective goals (goal attainment: items 3 and 4), be united and mutually trusting (integration: items 5 and 6), and transfer knowledge and values to employees (latency: items 7 and 8) (see Table 1). The items of the scale have been previously used in a study on digital leadership [34,35]. The internal consistency and reliability of the scale is good (Cronbach’s Alpha: t_1_ = 0.90 and t_2_ = 0.89).

PCC was measured using a scale consisting of ten items about the long-term care facilities’ leaders’ perceptions of the person-centeredness of the facility they lead. The scale includes three different elements of person-centeredness: (1) the degree of shared decision making, (2) care recipient participation, and (3) orientation toward client preferences (see Table 2). Parts of the scale were developed and cognitively pretested in a previous study of health care organizations [36]. The scale has been adapted to the context of LTCOs. Internal consistency of the scale is good (Cronbach´s Alpha: t_1_ = 0.89; t_2_ = 0.87). An additive score of the PCC items was created and standardized on a scale from 0 to 100 for the analyses, with higher values indicating greater levels of person-centeredness. The type of LTCO (stationary hospice, ambulatory hospice, ambulatory nursing care, and stationary care) and the survey cycle were included as control variables.

### 2.3. Statistical Analysis

We performed descriptive analyses and multivariate pooled cross-sectional regression analyses using Stata V.16.0, and the multivariate analyses tested whether agency capacity as measured by the AGIL scale, the survey cycle, interaction of these variables, and the type of facility predicted perceived PCC [37]. Within the regression analyses, we tested whether the beta coefficient of an independent variable is significantly different from zero. The beta coefficient predicts the degree of change in the outcome variable for every 1-unit of change in the independent variable.

## 3. Results

For the first survey cycle of 4333 eligible managers, 765 participated in the survey, of whom 503 fully and 207 partly completed the survey, and 25 did not agree to be interviewed. For the second survey cycle of 4185 eligible managers, 520 participated in the survey, of whom 294 fully and 192 partly completed the survey, and 29 did not agree to be interviewed. The analytical sample consisted of 503 managers in the first survey cycle and 294 managers in the second survey cycle after the exclusion of cases with missing information (Figure 1).

The analysis of the type of organization indicated that both surveys were comparable with regard to the types of organizations that participated (t test for mean level differences is not significant).

The perceived PCC score increased significantly between the outbreak of the pandemic and the second wave (*p* < 0.001) from 74 points during the first wave to 79 points during the second. Similarly, the mean scores on the AGIL scale increased significantly (*p* < 0.001) between the two survey cycles from 73 to 77 (Table 3). 

The multivariate regression analysis that included all variables simultaneously in model 1 showed that the AGIL score was significantly associated with the PCC score (Table 4). A 1% increase in agency capacity measured on the AGIL scale (range = 0–100) was associated with a 0.5% increase on the PCC scale score (range = 0–100). The survey cycle had a significant effect, indicating that institutional person-centeredness increased from the first pandemic wave (t1) to the second pandemic wave (t2; see model 1 in Table 4). The nonsignificant interaction term in model 2 between the survey cycle and the AGIL scale indicates that the association between perceived agency capacity and perceived PPC did not vary significantly by survey cycle (see also Figure 2). Accordingly, as shown in Figure 2, predicted margins of PCC increased with higher levels of AGIL but did not differ significantly between the two survey cycles. The introduction of organization type as a control variable showed that hospice, outpatient nursing care, and outpatient care facilities provided significantly higher PCC than inpatient care organizations (reference category), with the largest difference found between hospices and inpatient care organizations. 

## 4. Discussion

The main objective of this study was to test the hypothesis that the ability and power of LTCO to act flexibly, sustainably, and in a goal-oriented manner as a cohesive collective system, called systemic agency capacity, is a precondition for the ability to maintain high-quality routine tasks such as PCC during the COVID-19 pandemic. The regression results showed a significant association between perceived systemic agency capacity as measured by the AGIL scale and perceived PCC during the COVID-19 pandemic. This association did not differ between the first and second survey cycles, indicating that the association was quite stable over nine months. This significant association could be interpreted as preliminary proof for the hypothesis that the systemic agency capacity of nursing homes enables the provision of high-quality routine care during a pandemic. However, this is not a proof of causality but rather suggests that we could explore this hypothesis further. 

The results of this long-term care facility study are in line with the results of two other organizational studies [24,25,27]. These studies showed that hospitals that fulfill two of the four AGIL functions—goal orientation and integration—have higher implementation rates with regard to quality management or clinical risk management than hospitals that do not fulfill these two functions properly. As mentioned above, these two components are part of the systemic agency capacity construct. Additionally, there are some empirical hints in the literature regarding the importance of organizational properties such as social cohesion (integration function) and goal-oriented leadership (goal attainment function) for the promotion of PCC [10,11,31] that support our hypotheses and the result of this study. 

In addition, there are studies showing that in the context of the COVID-19 pandemic, social support—an important subdimension of social integration—is a useful resource for health care workers in coping with traumatic stress [38]. Social support was found to be related to stress resilience during the pandemic [39]. In times of the COVID-19 pandemic, social support can play an important role in maintaining the health of health care workers [40], for example, as a protective factor [41]. This is an important prerequisite for long-term care workers to provide high-quality care during a pandemic. 

Despite the supporting literature, these results should be interpreted with caution. One cannot exclude the possibility of a selection bias caused by the repeated cross-sectional design, which produced different participation rates in the survey cycles, and time constraints of top managers of nursing care facilities during the first and second waves of the pandemic. Leaders who felt more affected by the pandemic might have been more motivated to participate, which could have led to a selection bias. Although the study involved LTCOs throughout Germany, the results may not be representative of all LTCOs in Germany. For reasons of anonymization, the present study was not a panel study. In future studies, researchers should attempt to overcome data protection issues and design a panel study. Future studies could also include specific resilience resources such as pandemic-specific budgets to study their interaction with the general resilience resource agency capacity. Future studies might also consider measuring PCC from the patients’ perspective to counteract common method bias. 

## 5. Conclusions

LTCOs have had to face two challenges during the COVID-19 pandemic: coping with the pandemic and ensuring high-quality routine tasks such as PCC. We hypothesized that high systemic agency capacity as measured by the AGIL scale is a general core resource that enables long-term care facilities to maintain routine tasks such as delivering PCC despite the disruption caused by the COVID-19 pandemic. Our study provided some empirical support for this hypothesis. Owing to the constraints of this study, especially the repeated cross-sectional study design, these results are only preliminary. Despite these limitations, the results highlight the need for nursing leaders to focus attention on the social infrastructure for crisis management [29]. Maintaining the basic functions of an organization is a crucial but often neglected central task of nursing leaders. The main strategy should be to foster the systemic agency capacity of LTCOs by strengthening the central components of this agency capacity. Therefore, nursing leaders should implement structures and processes that (a) enable adaptability and efficacy through adequate tools [42]; (b) enable a common goal setting by consensus workshops and goal attainment by controlling the goal attainment progress with appropriate dashboards and controlling tools [25]; (c) enable solidarity by stressing the importance of “we” and developing a climate of supportive and cohesive relationships; and (d) enable knowledge and value transfer by establishing appropriate measures such as mentoring systems, onboarding events, or standard operation procedures [43,44,45,46].

## Figures and Tables

**Figure 1 ijerph-19-05045-f001:**
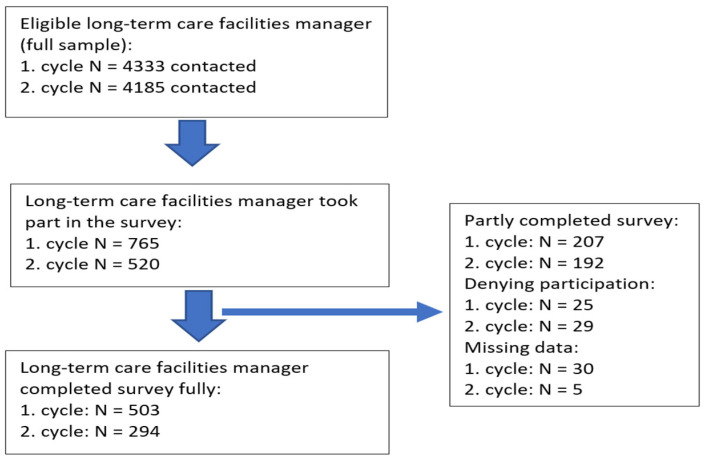
Flow chart of survey participation.

**Figure 2 ijerph-19-05045-f002:**
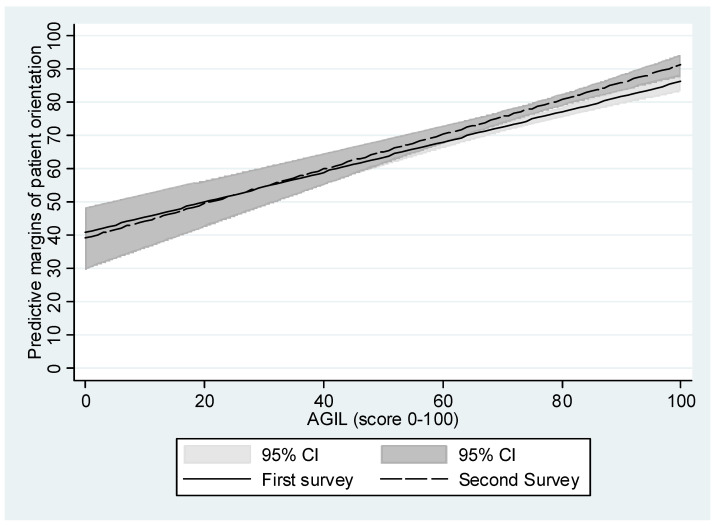
Association between perceived person-centered care (“patient orientation”) and AGIL scores by survey cycle.

**Table 1 ijerph-19-05045-t001:** Items of the AGIL scale surveyed among leaders of long-term care facilities in Germany during the COVID-19 pandemic.

	Thinking about your care facility in general, how strongly do you agree with the following statements?
1	We are very capable of adapting to changes in our environment.
2	In my area of responsibility, the business processes are highly effective.
3	It is easy for us to define important targets.
4	We pursue the defined targets with extraordinary persistence.
5	In our care facility, there is unity and agreement.
6	In our care facility, we trust one another.
7	We have excellent knowledge management.
8	We feel it is very important for new employees to internalize the values and attitudes of our care facility.

**Table 2 ijerph-19-05045-t002:** Items of the person-centered care (PCC) scale surveyed among leaders of long-term care facilities in Germany during the COVID-19 pandemic.

	Thinking about your care facility, how strongly do you agree with the following statements currently (since the outbreak of the COVID-19 pandemic)?
1	The care recipients are asked about their preferences regarding care.
2	The care provider and the care recipient jointly weigh the different care alternatives.
3	The care provider and the care recipient decide together which care will be provided.
4	The preferences of the care recipient are identified and explicitly taken into account during care.
5	Regular checks are made to see if care recipients still have questions.
6	The care recipient’s preferences regarding care are documented.
7	The care recipients receive company, support, stimulation, advice, encouragement and assistance in the process of change/adaptation.
8	The reference persons of care recipients receive support, guidance, advice, encouragement and assistance in the care situation.
9	At our care facility, we always adhere to standards and guidelines (e.g., treatment guidelines, care standards).
10	At our care facility, the care recipient’s relatives are involved in the care upon request of the care recipient.

**Table 3 ijerph-19-05045-t003:** Sample characteristics by survey cycle.

	First Survey Cycle		Second Survey Cycle		*t* Test for Mean-Level Differences
Total n (%)	503	(100%)	294	(100%)	
Perceived person-centered care ^a^ mean (sd)	74.3	(17.8)	79.3	(15.7)	*p* < 0.001
AGIL ^b^ mean (sd)	72.9	(13.7)	76.8	(13.6)	*p* < 0.001
Inpatient nursing care	110	(22%)	75	(25%)	*p* = 0.241
Hospice	17	(3%)	8	(3%)	*p* = 0.607
Outpatient nursing care	350	(70%)	202	(69%)	*p* = 0.797
Outpatient palliative nursing care	26	(5%)	9	(3%)	*p* = 0.162

Notes: ^a^ alpha score at t_0_: 0.89 and at t_1_: 0.87; ^b^ alpha score at t_0_: 0.90 and at t_1_: 0.88.

**Table 4 ijerph-19-05045-t004:** Association between AGIL as an independent variable and person-centered care as a dependent variable controlled for organization type and survey cycle.

	Model 1		Model 2	
	ß-coeff.	95% CI	ß-coeff.	95% CI
AGIL	0.478 ***	0.399–0.557	0.458 ***	0.359–0.556
Organization type (Reference: inpatient nursing care)				
Hospice	13.203 ***	6.820–19.587	13.265 ***	6.877–19.653
Outpatient nursing care	6.849 ***	4.265–9.433	6.852 ***	4.267–9.437
Outpatient palliative nursing care	1.496	−4.047–7.039	1.546	−4.001–7.092
Survey cycle	3.231 **	1.007–5.455	−1.011	−13.402–11.379
Interaction term				
Survey cycle × AGIL			0.056	−0.105–0.218
R^2^	0.2211		0.2211	
N	797		797	

Notes: ß-coeff., beta coefficient; CI, confidence interval. * *p* < 0.05, ** *p* < 0.01, *** *p* < 0.001.

## Data Availability

The data presented in this study are available on request from the corresponding author.

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
