# Peer review of "Is the Systemic Agency Capacity of Long-Term Care Organizations Enabling Person-Centered Care during the COVID-19 Pandemic? A Repeated Cross-Sectional Study of Organizational Resilience"

_ijerph, 2022, doi:10.3390/ijerph19095045_

Round 1

Reviewer 1 Report

The article examines the relationship between systemic agency capacity of long-term care organizations and perceptions of top managers regarding their organization’s ability to provide person-centered care. To me, it is not obvious what the term " systemic agency capacity" means. However, the concept is discussed and clarified carefully on page 2-3 – followed by a good introduction to why the concept is useful as a starting point to investigate to what degree long-term care organizations were able to provide person-centered care during the COVID-19 pandemic. Overall, I find that the paper is well written and well structured. The analysis is competently executed, and I think the paper brings in some new results.

Some minor remarks. The paper starts with section 1-3 and then starts with section 2 again on page 4 (followed by section 3-5). This needs to be fixed. I also encourage the authors to rewrite the abstract. As it stands now, it is difficult to follow - especially because the concept "system agency capacity" is used without being explained.

Author Response

Comment 1:   The article examines the relationship between systemic agency capacity of long-term care organizations and perceptions of top managers regarding their organization’s ability to provide person-centered care. To me, it is not obvious what the term " systemic agency capacity" means. However, the concept is discussed and clarified carefully on page 2-3 – followed by a good introduction to why the concept is useful as a starting point to investigate to what degree long-term care organizations were able to provide person-centered care during the COVID-19 pandemic. Overall, I find that the paper is well written and well structured. The analysis is competently executed, and I think the paper brings in some new results.

Response:       We would like to thank the reviewer for the positive evaluation of our manuscript.

Comment 2:     Some minor remarks. The paper starts with section 1-3 and then starts with section 2 again on page 4 (followed by section 3-5). This needs to be fixed.

Response:          We wanted to change the number of the chapters according your advice, but the second reviewer asked us to collapse chapter 1 to 3 into one introduction chapter. We had to follow his/her advice. Therefore, the numbers of the chapters changes, anyway.

Comment 3:     I also encourage the authors to rewrite the abstract. As it stands now, it is difficult to follow – especially because the concept "system agency capacity" is used without being explained.

Response:          We now changed the abstract in order to make it more comprehensible, including a short definition for the term "systemic agency capacity". Thank you for this important advice.

Reviewer 2 Report

Thanks for the opportunity to review. The manuscript is intriguing because unlike the other published papers, this one focuses on the managerial scenario. However, it is necessary to enrich and better define the study methodology for this reason I recommend a major revision

I suggest incorporating paragraph 2 and 3 introduction section, avoiding argumentative statements but providing a background and a lean rationale for the purpose of the study.

188 I suggest not to describe the sample, but to define the eligibility of the participants to whom the study is addressed

I suggest the adoption of the CROSS guidelines

https://www.equator-network.org/reporting-guidelines/a-consensus-based-checklist-for-reporting-of-survey-studies-cross/

195-197 these are results

198 This looks like a repeat of the design.

202 Again a repetition of the results that move to the appropriate section, perhaps using a flowchart

209 maybe give the name to the paragraph with measurements or outcomes

237 name the paragraph statistical analysis..

245 Timepoints must be defined in the methods not in the results

254 describe the method of multiple regression analysis in statistical analysis.. Was it hierarchical, sequential? Has the ΔR2 been calculated? what does βtesting refer to?

The results refer to the following concepts:

“a sequential multiple regression considering a criterion value (CV) and entering a different number of predictive variables (PV). Then, the goal of this sequential multiple regression is to estimate the individual contribution of each PV in modifying the variance CV, reported as variation in R2 (referred to as delta R2 or ΔR2)?” (ref: https://doi.org/10.3390/ijerph18189676 )

If so, enrich the statistical analysis section

277 Paraphrase the study objective as the beginning of the discussion and then describe the major findings of the study

Author Response

Comment 1:     Thanks for the opportunity to review. The manuscript is intriguing because unlike the other published papers, this one focuses on the managerial scenario. However, it is necessary to enrich and better define the study methodology for this reason I recommend a major revision.

Response:         We appreciate your comments. They have been very helpful. Thank you for that!

Comment 2:     I suggest incorporating paragraph 2 and 3 introduction section, avoiding argumentative statements but providing a background and a lean rationale for the purpose of the study.

Response:         As you recommended, we incorporated paragraph 2 and 3 into the introduction section and we rearranged the text substantially. By rearranging the manuscript, we tried to avoid too many argumentative statements and to provide a lean rationale for the purpose of the study (your recommendation).

Comment 3:     I suggest not to describe the sample, but to define the eligibility of the participants to whom the study is addressed.

Response:         As you recommended, we altered the section by describing the eligibility of the participants to whom the study is addressed.

Comment 4:     I suggest the adoption of the CROSS guidelines.

Response:         We now oriented at CROSS guideline. Thank you for this suggestion.

Comment 5:     195-197: This are results

Response:         In our opinion, this is a description of the sampling process.

Comment 6:     198:  "This looks like a repeat of the design"

Response:         Your statement is correct. Therefore, we skipped the text.

Comment 7:     202: Again a repetition of the results that move to the appropriate section, perhaps using a flowchart.

Response:          You are right. We now moved this description of the results to the result section and we now used a flowchart.

Comment 8:     209: maybe give the name to the paragraph with measurements

Response:          We now gave the paragraph the name "measures"

Comment 9:     237: name the paragraph statistical analysis

Response:          We followed your recommendation.

Comment 10:   245: Timepoints must be defined in the methods not in the results

Response:          Correct, we now canceled the description of the time points because they have been already described in the design section

Comment 11:   Describe the method of multiple regression analysis in statistical analysis. Was it hierarchical, sequential? Has the ΔR2 been calculated? what does βtesting refer to?

Response:          We used a pooled cross-sectional regression analysis that includes both cross-sectional survey waves simultaneously. A hierarchical regression analysis was not necessary because the data were cross-sectional surveys, not longitudinal. A Delta R square test was not needed because the significance test for the interaction term between survey wave and the independent variable is sufficient to test for differences in the association between the independent and dependent variable over time. The beta testing tests if the beta coefficient (b1, b2 ...) differs significantly from zero (Y = b1 x X1 + b2 x X2). If this is the case the factor tested could be interpreted as contributing to the variance of the dependent variable. We made some adjustments in the text.

Comment 12:   277: Paraphrase the study objective as the beginning of the discussion and then describe the major findings of the study.

Response:          We followed your recommendations. We placed now the study objective at the beginning of the discussion and presented then the results.

Round 2

Reviewer 2 Report

The manuscript has considerably increased the methodological character I can only suggest minor revisions:

L124: Between the design and the measures add the characteristics of the participants (inclusion and exclusion ..) as per CROSS guidelines.
L160: Add a description of beta-testing to the statistical analysis as in the answer to my first review .. "The beta testing tests if the beta coefficient (b1, b2 ...) differs significantly from zero (Y = b1 x X1 + b2 x X2). If this is the case the factor tested could be interpreted as contributing to the variance of the dependent variable. We made some adjustments in the text. "
L231 I suggest arguments of this type to enrich the panorama: "Thus, in the context of the ongoing pandemic, social support could be dominantly viewed as a useful option for an individual to rely on and use as emotional support to cope with traumatic stress. Evidence shows that social support is associated with resilience to stress and the reduction of depression and anxiety. Therefore, social support could play a key role in maintaining mental and physical health, not only in health workers but also in the general population " ref: https://pubmed.ncbi.nlm.nih.gov/29880040/ ; https://pubmed.ncbi.nlm.nih.gov/34574600/ 

Author Response

Thank you very much for your really helpful comments!

Changes: The changes are in italics (You will also see the changes in the manuscript in green colour)

Comment 1:   L124: Between the design and the measures add the characteristics of the participants (inclusion and exclusion ..) as per CROSS guidelines.

Response:         We appreciate your comment and we introduced a new paragraph and a new sentence as you recommended.

Selection of participants

The contact information (email addresses) for the German facilities was obtained from a freely accessible data register on the internet. We contacted 4,333 facilities by email, of which 3,195 were registered as outpatient care services, 865 as inpatient care services, and 273 as hospices (Hower et al. 2021; Pförtner et al. 2021). Only managers of a long-term care facility were eligible to participate in the survey. Other employees of a long-term care facility or managers of other facilities that do not provide long-term care were not allowed to participate in the survey.

Comment 2:     L160: Add a description of beta-testing to the statistical analysis as in the answer to my first review .. "The beta testing tests if the beta coefficient (b1, b2 ...) differs significantly from zero (Y = b1 x X1 + b2 x X2). If this is the case the factor tested could be interpreted as contributing to the variance of the dependent variable. We made some adjustments in the text. "

Response:         As you recommended, we incorporated a short description of beta-testing as follows at the last sentence of the paragraph you mentioned:

Statistical analysis

We performed descriptive analyses and multivariate pooled cross-sectional regression analyses using Stata V.16.0, and the multivariate analyses tested whether agency capacity as measured by the AGIL scale, the survey cycle, interaction of these variables, and the type of facility predicted perceived PCC (StataCorp, 2019). Within the regression analyses, we tested whether the beta coefficient of an independent variable is significantly different from zero. The beta coefficient predicts the degree of change in the outcome variable for every 1-unit of change in the independent variable.

Comment 3:     L231 I suggest arguments of this type to enrich the panorama: "Thus, in the context of the ongoing pandemic, social support could be dominantly viewed as a useful option for an individual to rely on and use as emotional support to cope with traumatic stress. Evidence shows that social support is associated with resilience to stress and the reduction of depression and anxiety. Therefore, social support could play a key role in maintaining mental and physical health, not only in health workers but also in the general population "

Response:         Thank you for this important comment. As you recommended, we added a new paragraph and additional literature to address your important point about the role of social support during the pandemic as follows:

In addition, there are studies showing that in the context of the ongoing COVID-19 pandemic, social support - an important subdimension of social cohesion and social integration - is a useful resource for health care workers in coping with traumatic stress (de Sire et al. 2021). Social support was found to be related to stress resilience during the pandemic (Hou et al. 2020). In times of the COVID-19 pandemic, social support can play an important role in maintaining the health of health care workers (Labrague 2021), for example, as a protective factor (Schug et al. 2021). This is an important prerequisite for long-term care workers to provide high-quality, person-centered care during a pandemic.